# The Acute Effects of Different Wearable Resistance Loads Attached to the Forearm on Throwing Kinematics, Myoelectric Activity and Performance in Experienced Female Handball Players

**DOI:** 10.3390/jfmk7030058

**Published:** 2022-08-17

**Authors:** Andrea Bao Fredriksen, Roland van den Tillaar

**Affiliations:** Department of Sports Sciences and Physical Education, Nord University, 7600 Levanger, Norway

**Keywords:** team-handball, overarm throwing, velocity, accuracy, resistance training, EMG

## Abstract

The purpose of this study was to investigate the acute effects of various loads attached to the forearm on throwing performance, kinematics and electromyography (EMG) activity in overarm throwing. A within-subjects design was used to evaluate thirteen experienced female handball players (age: 22.15 ± 2.82 years; height: 171.62 ± 7.68 cm; body mass: 73.35 ± 11.16 kg) who performed a penalty shot test with various loads attached to their forearms in three conditions: (1) no extra weight, (2) middle weight and (3) high weight. Performance together with 3D kinematics and EMG of eleven muscles were analyzed in Visual 3D (C-motion, Germantown, MD, USA) during the throw. The main findings were that peak velocity was affected (*p* = 0.004, η*p*^2^ = 0.37) but not accuracy (*p* = 0.47, η*p*^2^ = 0.06) when throwing with weights. However, there were no differences between the weights. Furthermore, EMG activity and most kinematics did not change with the added load; only the maximal angular elbow extension velocity increased (*p* ≤ 0.001, η*p*^2^ = 0.67), while the internal shoulder rotation velocity decreased with the load attached. We concluded that changes in throwing velocity were caused by the decrease in maximal angular internal shoulder rotation velocity. The increased maximal elbow extension velocity was probably caused by the increased moment of inertia of the forearm. Between mid and high weights, the load difference was probably too small to observe changes in kinematics due to the lower moment of inertia compared with studies that used heavier balls.

## 1. Introduction

Overarm throwing is an important part of sports such as baseball, javelin throw and handball, and maximal velocity and accuracy are important factors for performance [1]. Improving throwing technique through the timing of consecutive actions of body segments [2] and arm muscle strength might be expected to increase ball velocity. The most specific training for overarm throwing is practice of the technique. Overload can be achieved in two different ways: by increasing the duration of the load by varying the number of repetitions, or by using extra load on the forearm [2,3].

The overarm throw is a complex, discrete and fast movement [4]. Earlier studies have examined the effects of different types of training to enhance ball release velocity [5]. In most of the earlier studies, only ball release velocity was measured before and after a training period to verify the efficiency of the resistance training regime [6]. Some studies have investigated the effects of bench press or other strength tests to explain the improvement of ball release velocity due to increased strength in some muscle groups [3,7]. However, it is not known if or how these strength changes help increase ball release velocity. It is difficult to assess which factors cause the positive effect of the strength changes or training form and why [5].

The kinematics of overarm throwing in team handball have been examined in several studies [1,4,5,8]. The changes in throwing performance after the different types of training were probably caused by changes in the maximal angular velocity of the internal shoulder rotation and maximal elbow extension [1,5]. In earlier studies on well-trained players, it was reported that no differences in the maximal angles occur, but the timing was different between the ball weights when training with heavier balls, which had a negative effect on peak ball velocity, and the total moving time increased significantly [1,3,5].

A specific training method for overarm throwing is to use wearable resistance of the upper limb during throwing training. The external load applied on the forearms of the subjects is approximately 50–100% heavier than a normal ball (325–375 g). The principle for this training method is to improve strength and neural activation by the stimulus from the additional load, and simultaneously not adversely affect the specific movement pattern [9,10]. However, which exact mechanisms are causing performance improvement are not clear. In general, it is claimed to fulfill the desirable goal that the central nervous system compensates for the increased inertia by modifying the characteristics of muscle activation [11,12].

This training method offers some advantages. It maintains the specificity of the training, and training with wearable resistance, unlike training with heavier balls [1,3,5], permits individuality in the loading and the load can be applied for longer periods, such as after the player has thrown the ball and does not possess the ball [13].

To our best knowledge, there is only one study on training with external loads attached to the upper limbs in handball throwing. Skoufas et al. [13] investigated the effect of arm and forearm loading on novice handball players in a training intervention. The study investigated throwing velocity with and without external weights during the fifth and tenth weeks in the training program, followed by tests in the fifth and tenth weeks in the detraining period to investigate the maintenance of training adaptions that occur. The results for this study showed that ball velocity with and without external weights differed during the experiment. In the fifth week of the training period and the tenth week of the detraining period, the ball velocity was significantly lower during throwing with external weights than without [13]. In the tenth week of the training period and fifth week of the detraining period, the difference between the conditions (with and without external load) was minimal. The study showed the difference between the two conditions, but it did not investigate if the velocity with external loads increases throwing velocity without external loads. Moreover, the study investigated novice handball players, so the results might be misleading because their technique could have improved during the study due to a learning effect and not a training effect. Therefore, it would be interesting to look at how external load would affect experienced handball players. Skoufas et al. [13] did not investigate the acute effect. Thus, the aim of this study was to investigate the acute effects of different external loads attached to the forearm on overarm throwing performance, kinematics and electromyography (EMG) in experienced female handball players. It was hypothesized that the ball velocity would decrease with increasing external loads attached to the forearm, as found in earlier studies with heavier balls caused by lower maximal angular internal shoulder and elbow extension velocities [1,3,5], while EMG activity would be the same. Knowledge from this study can help athletes and coaches gain more insight about the effects of using wearable resistance in throwing and perhaps help them in targeting overarm training more effectively.

## 2. Methods

### 2.1. Design

To investigate the acute effects of different external loads on overarm throwing kinematics and performance in experienced female handball players, a within-subjects, repeated measures design was used. Three conditions with different weights attached to the forearm were tested, which were individualized and corrected by 50 g based on the participant’s body mass (Table 1). An earlier study by [5] showed that increasing ball weights by 100% showed a difference in absolute timing but not relative timing, so the heaviest weight that we used was approximately 100% heavier than a regular handball (325–375 g).

### 2.2. Subjects

Thirteen experienced female handball players participated in this study (age: 22.15 ± 2.8 years; height: 171.62 ± 7.7 cm; body mass: 73.35 ± 11.2 kg). They participated in competitions in the highest divisions in Norway. The subjects were fully informed about the protocol before participating in this study. Written consent was obtained prior to all testing from all subjects, with approval from the Norwegian Centre for Research Data (project number 182653) and in accordance with the current ethical standards in sports and exercise research.

### 2.3. Procedure

After an individual warm-up of 15 min, which consisted of jogging, throwing drills to warm up the throwing arm were conducted. Before the throwing drills, electrodes for EMG measurements, reflective markers and the forearm sleeve were attached to the participants to familiarize them with throwing while wearing the equipment. Throwing performance was tested in a penalty throw situation, which is a standing overarm throw toward a target 7 m away, always keeping the front foot on the ground. Everything was mirrored for the left-handers. The participants were instructed to throw each weight (0 kg, mid weight and high weight) as hard as possible and try to hit the target (Figure 1). Additional weights were attached to the forearm (Figure 2) on an exogen forearm sleeve (Lila™, Kuala Lumpur, Malaysia). The subjects had to aim at a circle target (0.6 × 0.6 m) at a height of 1.5 m, located in the upper middle of a handball goal (2 × 3 m). Ten attempts with each weight were recorded to measure throwing performance (ball velocity and accuracy), and the order of the conditions was randomized to avoid effects of learning and fatigue. Peak ball velocity was measured using a radar gun (Stalker ATS II, Richardson, TX, USA) in km/h. Throwing accuracy was measured with a video camera (Sony PXW-Z90V, Sony, Tokyo, Japan). The accuracy was analyzed using an x- and y-axis to measure the hit. The center was 20 cm in diameter, the next circle was 40 cm in diameter and the largest was 60 cm. Starting with the inner circle, throws were scored as 1, 2 and 3, and if they missed the target, the throw was scored as 4. The average in peak ball velocity and accuracy of all throws in each condition was calculated. Between each throw, the participants were allowed approximately 1 min of rest to avoid fatigue.

### 2.4. Measurement

A three-dimensional motion capture system (Qualisys, Gothenburg, Sweden) was used to measure the position of the reflective markers (2.6 cm in diameter) at a sampling rate of 240 Hz on the following anatomical landmarks: (a) foot: first and fifth phalanxes of the opposite foot of the throwing arm; (b) ankle: medial and lateral malleoli; (c) knee: medial and lateral epicondyles; (d) pelvis: anterior superior iliac spine; (e) hip: trochanter major on both sides; (f) shoulder: lateral tip of the acromion on both sides; (g) thorax: sternum; (h) elbow: medial and lateral epicondyles of the throwing arm; (i) wrist: styloid processes of the ulna and radius of the throwing arm; (j) hand: middle metacarpal head; (k) finger: middle distal phalanx; (l) ball: on top of the ball, left and right.

A Trigno Research+ System (Delsys, Natick, MA, USA) was synchronized with Qualisys and used to record EMG activity of the following muscles on the dominant side: deltoideus medius, deltoideus posterior, deltoideus anterior, trapezius transversalis, biceps brachii, triceps brachii long head, triceps brachii lateral head, pectoralis major, lattisimus dorsi, supraspinatus and serratus anterior. SENIAM recommendations were used for placements for the EMG [14]. The participant’s skin was shaved, scrubbed in alcohol, and dried with paper to reduce skin impedance before electrodes (27 mm × 37 mm × 13 mm, 14 g) were attached. Conductive gel (SignaGel, Parker Laboratories INC, Fairfield, NJ, USA) was applied to the electrodes to reduce noise. The sampling rate was at 1000 Hz.

Motion capture data and EMG data were exported to C3D files for segment modelling and analysis in Visual 3D software (C-motion, Germantown, MD, USA). Raw EMG signals were amplified and filtered with a preamplifier. These signals were high-pass and low-pass (500 and 20 Hz) filtered. Then, the signals were converted to root mean square (RMS) signals with a hardware circuit network, which had a common rejection rate of 106 dB. The throw was divided into three phases: (1) arm cocking phase, (2) arm acceleration phase and (3) follow-through phase [4]. Arm cocking started from onset of the wrist movement, from holding the ball with two hands, by moving the ball and upper extremity backward, while the hip started to move forward and rotate, and ended when the ball started to move forward. The next phase began when the ball moved forward after the maximal external rotation of the shoulder and ended with ball release (arm acceleration). The follow-through phase ended with the maximal internal shoulder rotation angle. The RMS means for each of the phases were calculated.

For motion capture data, all computations from the model-based data were smoothed with a low-pass Butterworth filter at a cut-off frequency at 15 Hz. The angle at ball release, maximal angle, maximal angular velocity and timing of maximal joint angles and joint velocity were calculated for the shoulder, elbow and wrist (Figure 3). The events’ joint angle and joint angular velocity were calculated in the distal to proximal orientation with a Cardan sequence in the order x-y-z. Timing was measured as time before ball release.

### 2.5. Statistics

Descriptive statistics were presented as means and standard deviations. Data were checked for normal distribution using the Shapiro–Wilk test. The various loads of weights on the forearm were analyzed using one-way ANOVA with repeated measures to compare the effects of different loadings on the forearm on velocity, accuracy, maximal joint angle, angle at ball release and maximal joint angular velocities. For EMG, a repeated 3 (conditions: no weight, mid weight, high weight) x 3 (phases: cocking phase, acceleration phase, follow-through phase) two-way ANOVA with repeated measures was performed. When significant differences were observed, a post hoc test using the Holm–Bonferroni correction was applied. If sphericity were violated, results with Greenhouse–Geisser corrections were reported. A significance level of 0.05 was used to identify differences. Effect size (ES) was evaluated with η*p*^2^ (ETA partial squared), where <0.01–0.06 constitutes a small effect, <0.06–0.14 constitutes medium effect and >0.14 constitutes a large effect [15]. The statistical analyses were conducted in SPSS version 27.0 (IBM Corp., Armonk, NY, USA).

## 3. Results

### 3.1. Velocity and Accuracy

A significant effect of attached weight was found for velocity (F = 6.9, *p* ≤ 0.004, η*p*^2^ = 0.37) but not accuracy (F = 0.7, *p* ≤ 0.47, η*p*^2^ = 0.06, Figure 4). Post hoc comparisons revealed that throwing velocity was significantly higher when throwing without weights compared to the other two conditions (*p* < 0.05).

### 3.2. Kinematics

No significant differences were found between the three conditions in maximal angles (F ≤ 2.05, *p* ≥ 0.15, η*p*^2^ ≤ 0.15), angles at ball release (F ≤ 2.14, *p* ≥ 0.143, η*p*^2^ ≤ 0.18) or timing of the maximal angles (F ≤ 1.3, *p* ≥ 0.2, η*p*^2^ ≤ 0.14, Table 2). A significant effect of condition was found for the maximal angular velocity in elbow extension (F = 21, *p* ≤ 0.001, η*p*^2^ = 0.67) and there was a nearly significant effect for the maximal angular velocity of the internal shoulder rotation (F = 2.8, *p* = 0.080, η*p*^2^ = 0.21). Post hoc comparison revealed that max angular velocity in elbow extension was significantly lower when throwing without weights compared to the other two conditions (*p* = 0.001), while a significant decrease was found between no weight and high weight for internal shoulder rotation velocity (*p* = 0.04).

### 3.3. EMG Activity

No statistically significant effects for the conditions (F ≤ 2.1, *p* ≥ 0.1, η*p*^2^ ≤ 0.16) or their interaction (F ≤ 1.4, *p* ≥ 0.24, η*p*^2^ ≤ 0.12) were observed. A statistical significance between phases was found in eight (anterior, medial and posterior deltoid, pectoralis major, serratus anterior, triceps brachii lateral head and triceps brachii long head) of the eleven muscles in the throw (F ≥ 3.1, *p* ≤ 0.046, η*p*^2^ ≥ 0.42, Figure 5).

## 4. Discussion

The aim of this study was to investigate the acute effects of throwing with various loads attached to the forearm on performance, kinematics and EMG. The main findings were that peak velocity was affected but not accuracy when throwing with weights. However, there were no differences between the weights. Furthermore, EMG activity and most kinematics did not change with the added load; only the maximal angular elbow extension velocity increased, while the internal shoulder rotation velocity decreased with the load attached.

The maximal ball velocity without external weights was in the same range (18 m/s) as found in earlier studies on experienced female handball players [1,7,16]. The external load on the forearm affected the maximal ball velocity, but no significant difference was found between mid and high weight. This was not in agreement with earlier studies on different ball weights [1,5]. When throwing with 20% differences in ball weights, [1] found a difference of 4.3% between maximal ball velocities compared to the regular ball, and they found a linear negative correlation with velocity and heavier ball weights. In the present study, the change between no weight and mid weight in ball velocity was 2.6%, and the change between no weight and high weight was 2.7%. These differences between the studies could be explained by the different locations of the loads. When throwing a weighted ball, the lever arm is larger than when the extra weight is attached to the forearm. The extra ball weight is thus much more sensitive to influence torque. This was found through the decrease in maximal angular internal shoulder rotation velocity between the no weight and high weight conditions (Table 3). Maximal angular velocity of internal shoulder rotation is one of the main contributors to a higher ball velocity [4], which decreased when throwing with heavier balls [1,5]. These findings explain the changes in the maximal ball velocity between no weight and the other two conditions, while the difference between mid and high weight was not large enough to result in a significant effect on the maximal internal shoulder velocities between the two conditions.

No significant differences in maximal angles nor changes in timing were found in this study, indicating that the attached load on the forearm did not change much of the kinematics. Only a significant and large effect was found for maximal angular elbow extension velocity, which increased with the attached weights on the forearm (Table 3). This was a clear discrepancy from earlier studies with heavier balls [1,5] that showed decreased maximal angular elbow extension velocity. This could be explained by the moment of inertia of weight placement (weighted balls vs. forearm). When throwing with weighted balls, the extra weight disappears when the ball leaves, while when throwing with external weight attached to the forearm, the moment of inertia increases during the whole throw, and it requires more force to slow down the movement [17]. Therefore, the elbow extension velocity can reach higher velocities with weight attached to the forearm.

Maximal EMG activity was found in accordance with an earlier study on female handball players [18]. Changes in the kinematics could be explained by the EMG activity. It was expected that the EMG activity between the conditions was higher with external weights, but no statistical significance differences between the conditions were found in the EMG activity. This could be explained by the instruction the subjects were given before they threw: “throw as hard as you can and try to hit the target.” When they were throwing as hard as possible on each throw, the subjects should have activated their motor units fully on each throw. Therefore, the activation was not expected to be higher in the other conditions. However, the effect size shows a large effect on three muscles between the conditions (biceps brachii, deltoideus anterior and supraspinatus, η*p*^2^ ≥ 0.14). These muscles all contribute to slow down movement. It suggests that external weights between conditions have a practical effect [19], but due to the low number of repetitions and durability, we cannot state this with high probability.

No significant differences in accuracy were found between the conditions. Several studies have investigated other factors, and these studies also showed that accuracy did not decrease. Van den Tillaar and Ettema [20,21] indicated that the type of instruction was important for throwing velocity by experts, but not for accuracy. Similar results were found between novice and expert players in a later study [22]. The findings in this study support these earlier findings, and it seems that external weights or any other factors that have been researched do not decrease accuracy. Therefore, it is beneficial to train with high throwing speed [23], even with extra loads attached to the forearm, because handball players can throw near maximum (80–90%) throwing velocity without decreasing their accuracy [22,24].

Some limitations in this study are that not all joint movements (trunk and pelvis movements) were analyzed. However, the load was attached to the forearm, which generally only influences arm movement. Furthermore, earlier studies have shown that these movements do not contribute much to overarm throwing movement in handball [4] and are not influenced by different loads (ball weights) [1]. Another limitation is that only female handball players were tested, and therefore, the findings cannot yet be generalized to men. However, earlier studies that investigated differences in throwing performance between men and women showed that kinematics (timing and maximal angular velocities) are not different between the sexes [25]. Due to the longer levers in men, they were able to throw faster [8,25]. Therefore, applying loads to the forearm in men would probably result in similar adaptations.

The practical implications of using wearable resistance attached to the forearm are that by using these loads, athletes can target maximal elbow extension velocity practice during throwing training more than with regular throwing, while accuracy of throws is not influenced. Over time, this type of training could perhaps result in faster throws.

## 5. Conclusions

Based on the findings of the present study, it can be concluded that loads attached to the forearm during overarm throwing in handball affect throwing velocity but not accuracy, and that that the amount of load (mid vs. high) does not change the effect. The difference in throwing velocity was probably caused by the decrease in maximal angular internal shoulder rotation velocity, but surprisingly, maximal elbow extension velocity increased, which was probably caused by the increased moment of inertia of the forearm. The load difference between the mid and high weights was probably too small to observe changes in kinematics due to the lower moment of inertia compared with studies that used heavier balls. Therefore, future studies should include men and a larger range of loads attached to the upper arm and forearm to investigate their effects on throwing performance and kinematics. Furthermore, research involving a training intervention should be conducted to study the long-term effect of training with this forearm loading protocol on the throwing performance of handball players.

## Figures and Tables

**Figure 1 jfmk-07-00058-f001:**
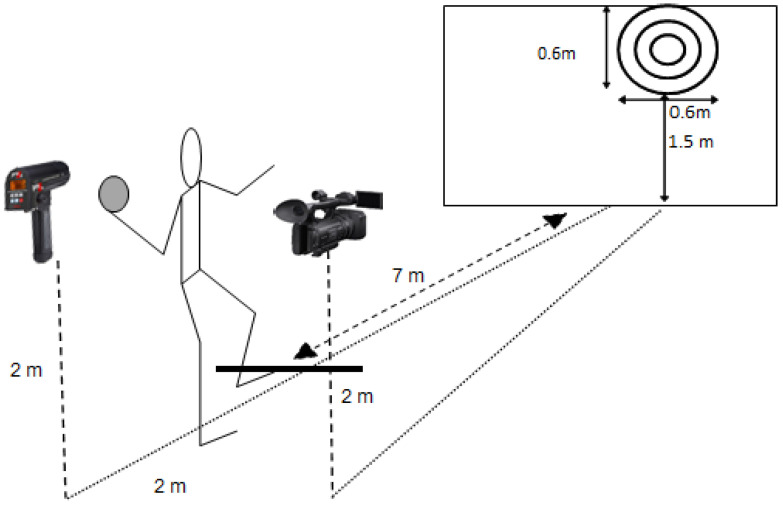
Experimental setup. Subjects stood 7 m from a target drawn on a large mattress. A radar gun measured ball velocity from behind the subjects.

**Figure 2 jfmk-07-00058-f002:**
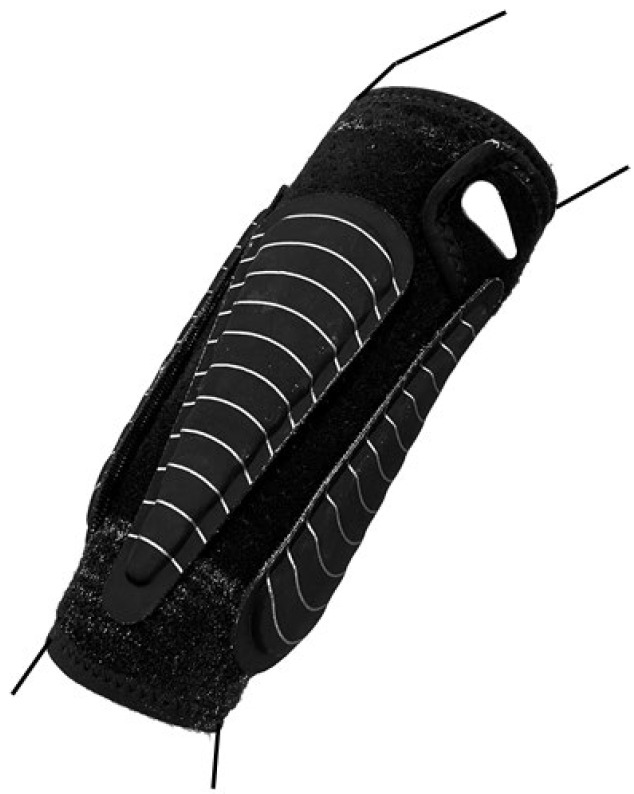
Attachment of the weights on the sleeve of the forearm.

**Figure 3 jfmk-07-00058-f003:**
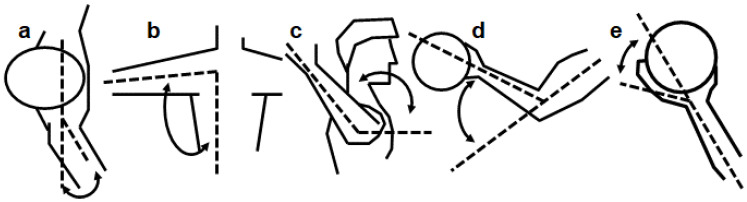
Definition of the kinematic parameters: (**a**) horizontal shoulder adduction, (**b**) shoulder abduction, (**c**) internal shoulder rotation, (**d**) elbow flexion and (**e**) wrist extension.

**Figure 4 jfmk-07-00058-f004:**
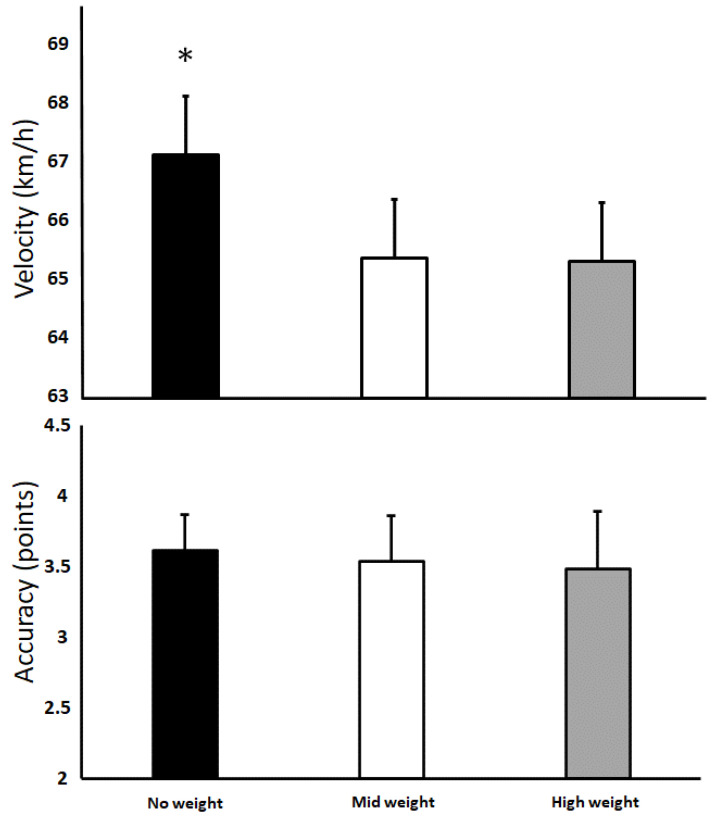
Mean values and standard deviations for maximal throwing velocity and accuracy in the three conditions. ∗ Indicates a significant difference from all other weights (*p* ≤ 0.05).

**Figure 5 jfmk-07-00058-f005:**
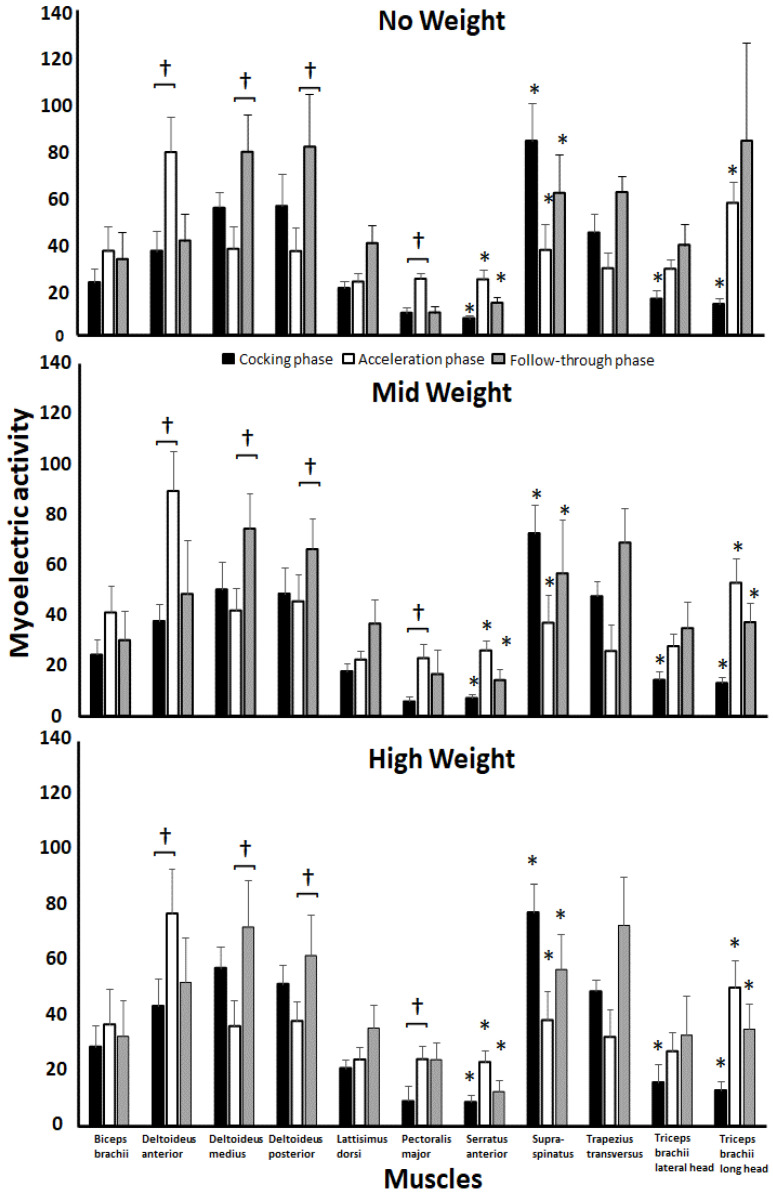
Mean values and standard deviations in EMG in the muscles with no, mid and high weight in the arm cocking, arm acceleration and follow-through phases. † Indicates significant difference between the marked phases (*p* ≤ 0.05). ∗ Indicates a significant difference between all of the other phases (*p* ≤ 0.05).

**Table 1 jfmk-07-00058-t001:** The different weights placed on the forearm in the three conditions based on the participant’s body mass.

Body Mass	No Weight (kg)	Mid Weight (kg)	High Weight (kg)
<60 kg	0	0.15	0.35
60–75 kg	0	0.20	0.40
76–90 kg	0	0.25	0.45
>90 kg	0	0.30	0.50

**Table 2 jfmk-07-00058-t002:** Angles at ball release and the maximal angles and their timings (mean ± SD) in all three conditions.

		Wrist Flexion/ Extension	Elbow Flexion	External/Internal Shoulder Rotation	Shoulder Horizontal Adduction	Shoulder Abduction
Angles at Ball Release (°)	No Weight	−7.8 ± 7.2	35.9 ± 8.0	69.7 ± 9.8	13.6 ± 9.5	84.4 ± 6.5
	Mid Weight	−5.5 ± 8.5	37.8 ± 3.8	77.0 ± 16.0	13.2 ± 9.3	90.5 ± 13.9
	High Weight	−8.7 ± 8.2	37.4 ± 6.0	74.0 ± 22.0	13.0 ± 8.0	86.7 ± 12.3
Maximal Angle (°)	No Weight	37.8 ± 12.4	118.0 ± 15.3	142.2 ± 20.2	−48.2 ± 29.2	-
	Mid Weight	34.3 ± 12.7	116.8 ± 16.8	143.6 ± 18.4	−48.6 ± 32.9	-
	High Weight	42.9 ± 16.3	116.3 ± 16.0	145.2 ± 19.0	−49.8 ± 34.5	-
Timing Max Angle (s)	No weight	−0.175 ± 0.138	−0.356 ± 0.231	−0.051 ± 0.045	−0.227 ± 0.159	-
	Mid Weight	−0.138 ± 0.117	−0.317 ± 0.251	−0.065 ± 0.017	−0.228 ± 0.185	-
	High Weight	−0.161 ± 0.217	−0.351 ± 0.247	−0.067 ± 0.013	−0.239 ± 0.147	-

**Table 3 jfmk-07-00058-t003:** Maximal velocities (mean ± SD) and their timings in all three conditions.

	Maximal Velocity			Timing Max Velocity (s)		
	No Weight	Mid Weight	High Weight	No Weight	Mid Weight	High Weight
Wrist flexion	988 ± 455	1159 ± 300	1474 ± 553	−0.010 ± 0.015	−0.007 ± 0.008	−0.008 ± 0.008
Elbow extension	644 ± 357 *	1190 ± 386	1218 ± 387	−0.021 ± 0.042	−0.019 ± 0.028	−0.028 ± 0.029
Shoulder horizontal adduction	447 ± 295	466 ± 326	499 ± 390	0	0	0
Internal rotation	3276 ± 687 †	3184 ± 749	3076 ± 834 †	0	0	0
Shoulder abduction	524 ± 258	464 ± 246	529 ± 327	0	0	0

* Indicates a significant difference between all conditions (*p* < 0.05). † Indicates significant difference between these conditions (*p* < 0.05).

## Data Availability

The raw data supporting the conclusion of this article will be made available by the authors, without undue reservation.

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
