# Peer review of "The Acute Effects of Different Wearable Resistance Loads Attached to the Forearm on Throwing Kinematics, Myoelectric Activity and Performance in Experienced Female Handball Players"

_jfmk, 2022, doi:10.3390/jfmk7030058_

Round 1
Reviewer 1 Report
First of all, I would like to congratulate the authors for their work. I will present some improvement suggestions that you should keep in mind.
Abstract
-Add the way the data was analyzed
-the main results should be presented (degree of significance and effect size)
Keywords
- Use keywords that are not in the title
Introduction
- I suggest that at the beginning of the introduction the research problem is clearly written.
-The literature review must be focused on the problem and it must be evidenced what the investigation adds to the knowledge
- According to the review, is it not possible to establish hypotheses?
Results
- I suggest that before each table and graphs what is shown is presented and after the tables and graphs the main results are presented.
- I suggest placing the significance value and effect size in the tables
Discussion
- At the end of the discussion, the practical implications of the investigation must be presented
Author Response
We want to thank the reviewer for revising the manuscript. We have now answered all questions to the reviewer and made the changes colored red in the manuscript. We think that it is now suitable for publication.
Reviewer 1
First of all, I would like to congratulate the authors for their work. I will present some improvement suggestions that you should keep in mind.
Abstract
-Add the way the data was analyzed
We have included this to the abstract now.
-the main results should be presented (degree of significance and effect size)
We have included this to the abstract now.
Keywords
- Use keywords that are not in the title
We have changed this now and included other key words.
Introduction
- I suggest that at the beginning of the introduction the research problem is clearly written.
Normally you end an introduction with the research question as we did. In the introduction you start with the specifying the area of research followed by what is done earlier in this area, and describe the evt. shortcomings and reasoning for the new research question. This structure we always use in our articles, which most readers like this structure. We have already introduced in the first paragraph the possibility of using extra loads. We hope the reviewer is satisfied with our explanation of the structure.
-The literature review must be focused on the problem and it must be evidenced what the investigation adds to the knowledge
As written in the introduction, not many studies have used extra weight on the forearms (to the best of our knowledge only one). That study did not investigate the acute effect and no kinematic or EMG analysis were performed, which are the main questions of this study to get more knowledge about what happens when throwing with wearable resistance. We have added practical reasoning on end of introduction for athletes and coaches.
- According to the review, is it not possible to establish hypotheses?
We have added an hypothesis at the end of the introduction.
Results
- I suggest that before each table and graphs what is shown is presented and after the tables and graphs the main results are presented.
We don’t want to us more sentences than necessary, that is why in the results we always started with the main results. Then the reader gets the main impression and wants to see this in graphs, which follows after each of the main results. We think that this is better for the reader to follow, then what the reviewer proposes. We hope the reviewer agrees with our point of view.
- I suggest placing the significance value and effect size in the tables
We had thought about it, but since most of the kinematics are not differently different (also with a low effect size, since it is not significantly different) and there are many numbers in the table. The extra numbers for level of significance and effects size for each of these variables would in our opinion not enhance reading of the tables and make the tables too large. Thereby, the readability of the tables worsens. We hope that the reviewer appreciates our point of view.
Discussion
- At the end of the discussion, the practical implications of the investigation must be presented
We have included this now to the end of the discussion.
Reviewer 2 Report
Extra period (after 'ing') line 63
It is unclear what is meant by "participants were asked to throw each weight in random order" (lines 113-114). How was the order randomized? Or was each participant able to decide which one to throw in what order?
Check format of Tables 2 and 3
In different parts of the manuscript, the authors use either EMG or myoelectric activity. I believe they are being use synonymously, but this should be clarified or one used consistently.
Is there any justification for why only females were included? I found this suprising when I got to the methods since sex differences were not mentioned in the introduction.
Author Response
We want to thank the reviewer for revising the manuscript. We have now answered all questions to the reviewer and made the changes colored red in the manuscript. We think that it is now suitable for publication.
Reviewer 2
Extra period (after 'ing') line 63
Changed now.
It is unclear what is meant by "participants were asked to throw each weight in random order" (lines 113-114). How was the order randomized? Or was each participant able to decide which one to throw in what order?
The order was counterbalanced randomized (6 different possible orders) each order was used on two subjects. This is now added to the text: and the order of the conditions was counterbalanced randomized by the scientist to avoid an effect of learning and fatigue.
Check format of Tables 2 and 3
We have changed the tables a little bit to increase readability.
In different parts of the manuscript, the authors use either EMG or myoelectric activity. I believe they are being use synonymously, but this should be clarified or one used consistently.
This is now all changed in EMG throughout the text.
Is there any justification for why only females were included? I found this surprising when I got to the methods since sex differences were not mentioned in the introduction.
The justification is that we had women available for our study. Would you ask the same question when we would have used only men? We have added female to the title to avoid this evt. confusion.
Reviewer 3 Report
The manuscript describes a study investigating the effects of added load to the forearm on performance and kinematics of female handball players in throwing a ball to a target. The paper is well written and present interesting results. I have only one major issue and a suggestion.
Major: The study seems under powered. A brief check on GPower shows that the sample size required for medium effects (80% of power) in a repeated measures ANOVA would be of, at least, 27 participants. The power (for the medium effect size), currently, is of 45%. I understand the issue of collecting data in expert participants, but the analyses, as they stand, can only deal with large effect sizes. I would suggest collecting more participants.
Suggestion: I would suggest to the authors to observe whether relative timing between body segments change provided the weight. This would refer to changes in overall movement pattern coordination - which can be positive for later improvements (see, for a long discussion on this, Kugler et al. (1982) - On the Control and Co-ordination of Naturally Developing Systems; or, for empirical attempts see Dan Southard papers on throwing).
Minor:
Line 63 - Can't the different ball's weights be individually selected also?
Line 63 - There is a "." between "training" and "and".
Lines 83-85 - What is the usefulness of studying the acute effects in this case? I think the authors should motivate such choice (e.g., the EMG refers to potential adaptations...)
Author Response
We want to thank the reviewer for revising the manuscript. We have now answered all questions to the reviewer and made the changes colored red in the manuscript. We think that it is now suitable for publication.
Reviewer 3
The manuscript describes a study investigating the effects of added load to the forearm on performance and kinematics of female handball players in throwing a ball to a target. The paper is well written and present interesting results. I have only one major issue and a suggestion.
Major: The study seems under powered. A brief check on GPower shows that the sample size required for medium effects (80% of power) in a repeated measures ANOVA would be of, at least, 27 participants. The power (for the medium effect size), currently, is of 45%. I understand the issue of collecting data in expert participants, but the analyses, as they stand, can only deal with large effect sizes. I would suggest collecting more participants.
|
|
In an earlier study with weighted balls we already observed significantly differences in kinematics and throwing performances when using only 7 subjects (van den Tillaar & Ettema, 2004) and using ball weights similar to the total loads attached to the forearm. Based on the formula for sample size calculation:
And the data from earlier study (van den Tillaar & Ettema, 2004) we only needed 6 subjects to have enough power to avoid underpowered. Therefore, we did not think that the study was under powered, with 12 subjects.
Suggestion: I would suggest to the authors to observe whether relative timing between body segments change provided the weight. This would refer to changes in overall movement pattern coordination - which can be positive for later improvements (see, for a long discussion on this, Kugler et al. (1982) - On the Control and Co-ordination of Naturally Developing Systems; or, for empirical attempts see Dan Southard papers on throwing).
We have done earlier studies with weighted balls where we also looked ad relative timing and did not find any differences. Thus, we would not expect that here either, when the absolute timing did not change much.
Minor:
Line 63 - Can't the different ball's weights be individually selected also?
We agree that this is possible, but during training all athletes throw with the same ball, while individually you can adapt the load attached to the forearm for each subject, when training together. This way the training is individualized. So practically easier to use during training. Furthermore, there are not many different weighted balls available in handball.
Line 63 - There is a "." between "training" and "and".
This is changed now.
Lines 83-85 - What is the usefulness of studying the acute effects in this case? I think the authors should motivate such choice (e.g., the EMG refers to potential adaptations...)
We have added some more implications and hypothesis to the end of the introduction for the choices.